# MixTrain: Accelerating DNN Training via Input Mixing

## Abstract

Training Deep Neural Networks (DNNs) places immense compute requirements on the underlying hardware platforms, expending large amounts of time and energy. An important factor contributing to the long training times is the increasing dataset complexity required to reach state-of-the-art performance in real-world applications. To address this challenge, we explore the use of input mixing, where multiple inputs are combined into a single composite input with an associated composite label for training. The goal is for training on the mixed input to achieve a similar effect as training separately on each the constituent inputs that it represents. This results in a lower number of inputs (or mini-batches) to be processed in each epoch, proportionally reducing training time.

We find that naive input mixing leads to a considerable drop in learning performance and model accuracy due to interference between the forward/backward propagation of the mixed inputs. We propose two strategies to address this challenge and realize training speedups from input mixing with minimal impact on accuracy. First, we reduce the impact of inter-input interference by exploiting the spatial separation between the features of the constituent inputs in the network's intermediate representations. We also adaptively vary the mixing ratio of constituent inputs based on their loss in previous epochs. Second, we propose heuristics to automatically identify the subset of the training dataset that is subject to mixing in each epoch. Across ResNets of varying depth, MobileNetV2 and two Vision Transformer networks, we obtain upto $1.6\times$ and $1.8\times$ speedups in training for the ImageNet and Cifar10 datasets, respectively, on an Nvidia RTX 2080Ti GPU, with negligible loss in classification accuracy.

## 1 Introduction

The success of deep neural networks has come at a cost of rapidly rising computational requirements for training. This increase is due to a combination of rising dataset and model complexities. For example, in the context of image classification, training dataset complexity increased significantly from MNIST and CIFAR-10/100 (50,000 - 60,000 images) to ImageNet-1K (1.2 million) and ImageNet-21K (14.2 million). This is supplemented by a growth in model complexity required to achieve state-of-the-art performance (Stojnic et al., 2023). The impact of increased training computation is both monetary (cost to train) and environmental ($CO_2$ emissions) (Strubell et al., 2019). A study from OpenAI (Amodei et al., 2018) reports that training costs of deep neural networks have been doubling every 3.5 months, greatly outpacing improvements in hardware capabilities.

**Prior Efforts on accelerating DNN Training**: Several methods have been proposed to accelerate DNN training. We divide them into a few broad categories, such as enabling the use of large-scale parallelism (e.g., hundreds or thousands of servers) in DNN training (You et al., 2017; Goyal et al., 2017), training on reduced-resolution inputs (Touvron et al., 2020; Tan & Le, 2021), training at reduced precision (Sun et al., 2019), pruning to reduce the model size during training (Lym et al., 2019), input instance skipping (Jiang et al., 2019; Zhang et al., 2019) and dataset condensation (Mirzasoleiman et al., 2020; Killamsetty et al., 2021).

mixTrain: **Accelerating DNN Training by mixing inputs**: Complementary to the aforementioned efforts, we propose the use of input mixing, a technique that has traditionally been used for data augmen-

tation (Zhang et al., 2017; Yun et al., 2019), to accelerate DNN training. Consider two training inputs $x_1$ and $x_2$. A mixing function F is applied to $x_1$ and $x_2$ to produce a *mixed input X*. The mixed input can be thought of as a point in the input space that combines information from both the constituent inputs that it represents. From the functional perspective, training on a mixed input must produce a similar effect on the model as training on the individual constituent inputs. On the other hand, from a computational viewpoint, mixing inputs reduces the number of input samples that need to be processed during training. This reduction in the effective size of the training dataset leads to fewer mini-batches in each epoch, and thereby lower training time. Due to the nature of input mixing, it is complementary to, and can be combined with, the other approaches to accelerate training described above. In mixTrain, we adopt computationally lightweight mixing operators CutMix and MixUp that have been proposed for a different purpose, *viz.* data augmentation (Zhang et al., 2017; Yun et al., 2019). As illustrated in Fig. 1, MixUp performs a simple weighted linear averaging of the pixels of two inputs, while CutMix randomly selects a patch of one input and pastes it onto the other.

Realizing training speedups through input mixing raises interesting questions, such as how to train networks on mixed samples, which samples to mix, etc. We observe that indiscriminate application of mixing leads to a considerable drop in learning performance and model accuracy. On further investigation, we find that this can be attributed to the interference between the processing of the constituent inputs within each mixed input. To preserve accuracy, we therefore propose techniques to mitigate this interference. We find that for the CutMix operator, the network's internal features largely maintain spatial separation between the constituent inputs in convolutional layers, but this separation is lost in the fully connected layers. We thus propose *split propagation*, wherein the features corresponding to each constituent input are processed separately by the fully connected layers. In contrast, with the MixUp operator, spatial separation between the constituent inputs is not maintained. Here, we mitigate the impact of interference through *adaptive mixing*, where the weights of the constituent inputs are varied based on their losses in previous epochs.

Additionally, we explore applying mixing selectively, i.e., only for a subset of training inputs in each epoch. We design a loss-driven metric to identify the training samples that are amenable to mixing in each epoch. We find that inputs at the two ends of the loss distribution, i.e., with very low and very high loss magnitudes, are amenable to mixing. Low-loss inputs are mixed because their functional performance remains largely unaffected by mixing. In contrast, we mix samples with high loss because a considerable percentage of such samples are unlikely to be learned even when no mixing is applied. We show that mixTrain achieves superior accuracy vs. efficiency tradeoffs compared to alternative approaches such as input skipping and early termination. Finally, we note that mixTrain is designed in a completely hyper-parameter free manner. This reduces the additional effort spent on hyper-parameter tuning for different models.

The key contributions of this work can be summarized as follows.

- To the best of our knowledge, mixTrain is the first effort to reduce the complexity of DNN training by mixing inputs

- We propose two strategies to improve the learning performance of mixTrain. First, we propose split propagation and adaptive mixing to reduce the impact of interference between the constituent inputs in a composite sample. Second, we apply mixing selectively, i.e., only on a subset of the training dataset every epoch.

- Across our benchmarks consisting of both image recognition CNNs (including ResNet18/34/50 and MobileNet) and vision transformers, we demonstrate up to $1.6\times$ and $1.8\times$ improvement in training time on the ImageNet and Cifar10 datasets respectively for $\sim$0.2% Top-1 accuracy loss on a Nvidia RTX 2080Ti GPU, without the use of additional hyper-parameters.

## 2 Input Mixing: Preliminaries

Input mixing takes multiple inputs and combines them into a composite input, taking in information from each of the constituent inputs. mixTrain uses two operators - MixUp (Zhang et al., 2017) and CutMix (Yun et al., 2019), which are illustrated in Fig. 1.

Consider two inputs, $x_1$ and $x_2$. For MixUp, as seen in Equation 1, each pixel $j$ of the composite input $X$ is obtained by linearly averaging the corresponding pixels of $x_1$ and $x_2$. The mixing ratio $r$ is in the range [0, 1]. The CutMix operator selects a random patch of $x_1$, and pastes it onto $x_2$. The weightage $r$ of each input $x_i$ is decided by its area in the composite sample.

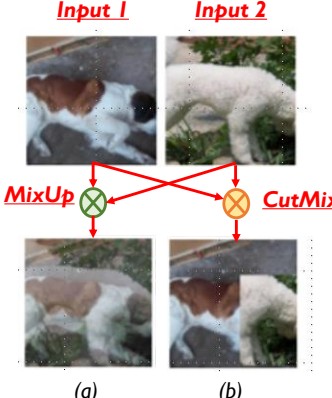

$$X_j = r \cdot x_{1,j} + (1 - r) \cdot x_{2,j} \tag{1}$$

Further, let us assume the target labels of the constituent inputs are $y_1$ and $y_2$. In (Zhang et al., 2017; Yun et al., 2019), the loss of the composite input $X$ is defined as the weighted sum of the loss of $X$ with respect to $y_1$ and $y_2$, as shown in Equation 2 for the cross-entropy loss. Here, $f$ is the DNN model, and $K$ the number of classes.

Input mixing has previously been applied for data augmentation, wherein randomly selected training input samples are combined through operators such as (Zhang et al., 2017; Yun et al., 2019) and added to the training set. Training on the randomly combined input samples has the effect of virtually augmenting the dataset, as the model is exposed to new training samples in each epoch. These efforts are focused on improving generalization, often

Figure 1: Mixing operators (a) MixUp (b) CutMix

achieved at the cost of increased training time. Specifically, the total number of input samples in each epoch of training after mixing remains the same. Further, in order to realize improvements in accuracy, these techniques often require 2-3× more training epochs than baseline SGD (Yun et al., 2019; Zhang et al., 2017).

$$Loss(X) = -(\alpha \cdot log(\frac{e^{f(X)_{y_1}}}{\sum_{l=1}^{K} e^{f(X)_l}}) + (1 - \alpha) \cdot log(\frac{e^{f(X)_{y_2}}}{\sum_{l=1}^{K} e^{f(X)_l}})) \tag{2}$$

## 3   mixTrain: **Accelerating DNN Training via Input Mixing**

The key idea in mixTrain is to improve the overall training time by dynamically applying the mixing operators, MixUp and CutMix, on the training dataset $D$ to reduce the number of samples in each epoch. However, naive mixing, e.g., where random pairs of input samples are mixed in each training epoch to reduce the number of training samples by half, negatively impacts classification accuracy. As observed in Fig. 2(a), on the ImageNet-ResNet50 benchmark, the drop in accuracy incurred after training on the reduced (i.e., halved) dataset obtained after applying either operator is nearly 4-6%.

The following subsections discuss the two key strategies that are critical to the overall success of mixTrain, namely, reducing the impact of interference between constituent inputs and selective mixing.

### 3.1   Reducing Impact of Interference

In this subsection, we discuss the primary cause affecting the accuracy of training with naive mixing, *i.e.*, interference between constituent inputs, and propose techniques to address the same.

We begin by analyzing the ability of a network trained with mixed inputs to correctly classify the constituent inputs of a composite sample. At different stages of training (different training epochs), we identify the set of training samples that the network classifies correctly without mixing, say set $S$. Our goal is to understand how the network fares in classifying the samples in set $S$ after they have been mixed. Specifically, we study the network's performance in detecting the presence of both constituent inputs in the mixed sample. Consider inputs $x_1$ and $x_2$ in $S$ mixed with ratio $\alpha = 0.5$ to form $X$, which is passed through the network. The network detects constituent inputs $x_1$ and $x_2$ in $X$, when the softmax scores of their corresponding class labels occupy the highest and second highest positions (order can be inter-changeable between $x_1$ and $x_2$). Only a single input is detected when the class label of one of the constituent inputs has the highest softmax score (say $x_1$), while the second-highest score is achieved by a class not corresponding to the second constituent input (i.e., other than $x_2$).

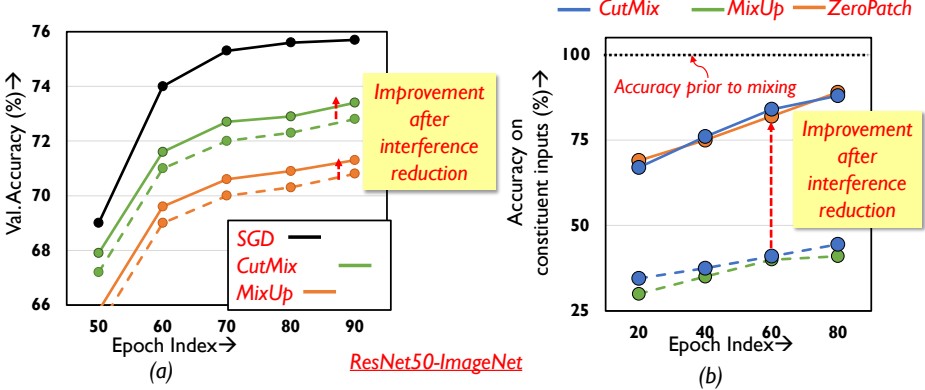

Figure 2: Classification performance with mixed inputs

Samples in set $S$ are thus mixed in pairs ($r = 0.5$), and the accuracy on the mixed inputs is recorded. Five such runs are conducted to allow for different random input combinations and the results are averaged and presented in Fig. 2. Surprisingly, after mixing is applied, the network is able to classify only less than half of the inputs in $S$ (green and blue dotted curves in Fig. 2(b)) even in the final epochs of training- note that these were inputs that were classified correctly without mixing (black line). On further investigation, it is found that for many mixed inputs, the network is able to correctly classify only one of the constituent inputs. The class label of the other constituent input often does not appear even amongst the Top-5 predictions made by the network. This leads to increased loss for one of the constituent samples, consequently impacting training performance and the final validation accuracy. It is thus critical to develop techniques that effectively learn on all constituent samples of a composite input. We next describe our approach to addressing this challenge.

**Split Propagation**: We identify two factors that contribute to the poor classification accuracy of a mixed input's constituent inputs in the case of the `CutMix` operator. Due to the random nature of the patch selected from a constituent input, it is possible to miss the corresponding constituent inputs' class object. Second, there may be interference between the features of the constituent inputs when the network processes the mixed sample. To design effective strategies that improve overall classification performance, it is important to understand the individual effect of each factor. We study the impact of the first factor by passing random patches from the inputs through the network; however, instead of mixing, random patches amounting to half the input area are zeroed-out. As shown using the solid orange curve (ZeroPatch) in Fig. 2(b), the drop in accuracy is only ~16%, and is significantly lower compared to mixing. This indicates that it is the interference between the constituent inputs that is the primary factor causing degradation in classification performance.

Examining the intermediate representations of the network while processing mixed inputs sheds some light on this interference. By virtue of the nature of convolutions, the spatial separation between constituent inputs in the composite input is maintained through many layers of the network, with only mild interference occurring at the boundaries of the inputs. For example, in Fig. 3, the right half of the features in the final convolution layer's output pertain to the right half of the mixed input. The spatial distinction between the features is maintained until the last convolutional layer, but is lost after the averaging action of the final pooling layer. As a result, the fully connected layer correctly classifies only one of the constituent inputs[1].

To aid the network in classifying both constituent inputs correctly, we propose split propagation of constituent features after the final convolution layer. As shown in Fig. 3, we identify the region in the final convolutional layer's output maps pertaining to each constituent input, and pass the features separately through the remaining layers of the network. Both constituent inputs of mixed samples are now classified correctly, leading to a significant improvement in classification performance (solid blue curve in Fig. 2(b)). During back-propagation, the output errors of each constituent input are propagated separately until the average pooling layer. The error tensors obtained at the input of the average pooling layer are then concatenated

---

[1](Zhang et al., 2017; Yun et al., 2019) resolve this issue by exposing the constituent inputs twice in each epoch through two different mixed inputs. While this improves accuracy, it defeats our objective of improving training runtime

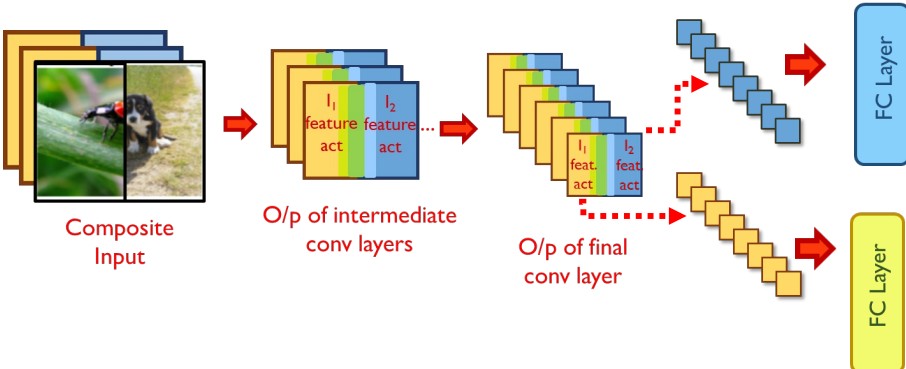

Figure 3: Training Mixed Inputs

and propagated backwards across the rest of the network. The classification loss for the constituent inputs improves, thereby improving overall validation accuracy (Fig. 2(a)). We note that the split propagation of the constituent inputs can be performed in parallel. Thus, the runtime overheads of this scheme are negligible, accounting for $< 3\%$ of overall training time.

**Adaptive Mixing**: Unlike `CutMix`, the `MixUp` operator averages each element of the constituent inputs prior to processing the network. Therefore, the network's internal representations do not exhibit any spatial separation between the constituent inputs. We thus devise alternative strategies to mitigate the impact of inter-input interference.

It appears from Fig. 2(a) that the validation accuracy with `MixUp` is even lower compared to `CutMix`, due to a slower rate at which training loss improves for the mixed inputs. Naturally, a simple boost in performance can be achieved by at least improving the loss for one of the constituent inputs of the mixed input. We thus adapt the weight ($r$) of constituent inputs so as to favour the more difficult input, as identified by the loss in the previous epoch. However, if the constituent samples were mixed in the previous epoch, it is not trivial to obtain their individual losses prior to mixing. To that end, we utilize an approximation to evaluate the losses of the constituent inputs in the previous epoch, described as follows. Consider two constituent inputs $x_1$ and $x_2$ with target labels $y_1$ and $y_2$ respectively, that have been mixed with ratio $r_E$ in epoch $E$ (Equation 3), to form the composite sample X. As seen in Equation 4, we use the loss of the network on the mixed input $X$ to estimate its loss on the individual constituent inputs. Here, $K$ stands for the number of classes in the task. While estimating the loss of $x_1$ and $x_2$ in such a manner is indeed an approximation, this allows us to avoid an additional forward propagation step to estimate the true loss of $x_1$ and $x_2$, thereby alleviating any runtime overhead.

$$X = r_E * x_1 + (1 - r_E) * x_2 \tag{3}$$

$$Loss(x_1, E) = -log(\frac{e^{f(X)_{y_1}}}{\sum_{l=1}^{K} e^{f(X)_l}}) \quad Loss(x_2, E) = -log(\frac{e^{f(X)_{y_2}}}{\sum_{l=1}^{K} e^{f(X)_l}}) \tag{4}$$

Once the losses of the constituent inputs have been obtained, we mix them in the next epoch $E+1$ with the ratio $r_{E+1}$ as shown below. As seen in Fig. 2(a), this provides a boost in classification accuracy.

$$r_{E+1} = \frac{Loss(x_1, E)}{Loss(x_2, E)} \tag{5}$$

Note that there is still some gap between the accuracy with and without mixing even after the use of split propagation and adaptive mixing, which we address next.

## 3.2   Selective Mixing

We explore a second strategy, selective mixing, to further improve accuracy when training with mixed inputs. Here, the general principle is to dynamically identify a subset of the training dataset in each epoch for which mixing does not have a negative impact on overall classification performance. We achieve this through the design of a loss-based amenability metric that determines, for each epoch, the subset of samples $S_{mix}$ that can be mixed in subsequent epochs. Samples that are not amenable to mixing are added to set $S_{noMix}$. The training dataset is thus formed using samples in $S_{noMix}$ as is, and mixing pairs of samples in $S_{mix}$.

**Overview**: The proposed selective mixing strategy consists of three steps as shown in Fig. 4. At every epoch, the reduced dataset is divided into mini-batches and fed to the network. The network performs the forward and backward passes on each mini-batch. Once the forward pass for a particular mini-batch is complete, the loss of each constituent input is computed. This is used to determine the amenability of each constituent input to mixing in the next epoch E+1, subsequent to which it is added appropriately to $S_{mix}$ or $S_{noMix}$. Finally, the batch-sampler forms mini-batches for the epoch E+1 by randomly drawing samples from either $S_{mix}$ or $S_{noMix}$.

The first and the third steps are straight-forward. In the following sub-section, we elaborate on the second step, i.e., determining the amenability of a sample to mixing, in greater detail.

### 3.2.1   Evaluating amenability to mixing in Epoch E

A suitable loss-based metric must estimate the subsets $S_{mix}$ and $S_{noMix}$ every epoch, such that no negative impact on accuracy is suffered. We design such a metric by studying trends in the loss of a sample prior to and after mixing, at different stages of the training process.

Consider models trained with `MixUp` and `CutMix` at three different training epochs as shown. At each selected epoch, we compute the $L_1$ difference of the loss of every sample $x$ with and without mixing, i.e., $loss_{mix}(x)$ and $loss(x)$ respectively. We define $loss_{mix}(x)$ as the loss of the mixed sample $x'$ with respect to the golden label of $x$, as shown in Equation 6. Here, $K$ is the number of classes, and $y$ is the golden label of $x$. We average $loss_{mix}(x)$ after 5 different random pairings to create $x'$.

$$Loss_{mix}(x) = -log\left(\frac{e^{f(x')_y}}{\sum_{l=1}^{K} e^{f(x')_l}}\right) \tag{6}$$

We observe that $loss_{mix}(x)$ deviates and increases further away as $loss(x)$ increases, consistently across the benchmarks analyzed for both operators (Fig. 5(a) depicts the same for `CutMix`). In other words, the graph indicates that *as loss(x) increases, its amenability to mixing decreases*. Furthermore, we find that prior to mixing, a majority of the correctly classified samples occupy the low loss regime as shown in Figure 5(a).

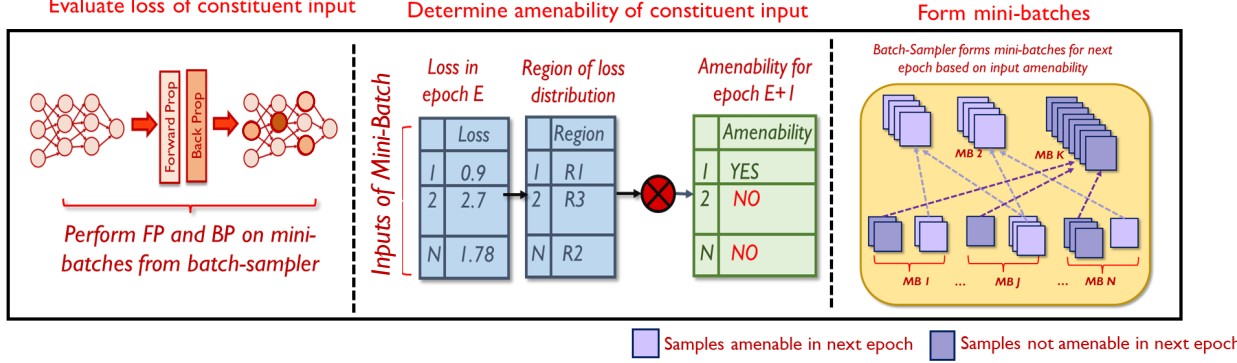

Figure 4: Overview of Selective Mixing

After applying mixing to these samples, we find that their classification accuracy is largely retained, especially as epochs progress, as depicted in Fig. 5(b) for the `CutMix` operator.

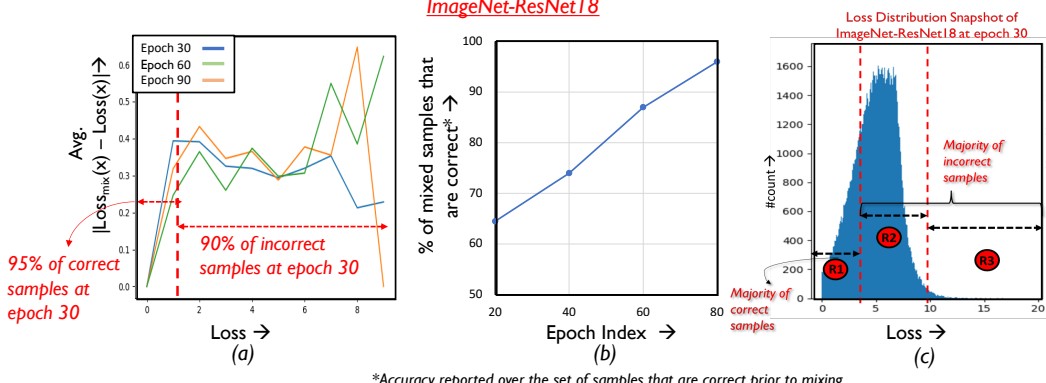

Figure 5: Analyzing amenability to mixing

Hence, for samples that are not mixed in epoch E, we determine their amenability to mixing in the next epoch based on the particular region of the loss distribution it belongs to. As illustrated in Fig. 5(c), the loss distribution is divided into three regions that utilize a different criteria for gauging amenability. We now discuss the criteria for each region, and the conditions for continuing mixing in subsequent epochs.

Region 1 corresponds to the area in the loss distribution where a majority of the correctly classified samples are located. From Fig. 5(b) we know that the loss, and to a certain extent the classification accuracy of such samples remains largely unaffected by mixing and are hence mixed aggressively. Next, we consider the portion of the loss distribution occupied by the incorrect samples and divide this space into two regions. Region 2 comprises of incorrect samples with moderate loss. To avoid any negative impact on accuracy, we avoid mixing these samples. Moving on to Region 3, these are samples the network finds very difficult to classify as characterized by their high loss magnitudes. We find that the training effort can be reduced on samples that consistently occur in Region 3 by mixing them, as they are unlikely to contribute to final classification accuracy.

The separations in the loss distribution are realized using simple linear clustering techniques that correlate the loss of a training sample in some epoch E to classification accuracy, based on trends in previous epochs. Let $L_{corr}$ and $L_{incorr}$ represent the running average of the correct and incorrect samples in $S_{noMix}$ respectively (calculated from epoch 0 to E-1), and let $L_{mid}$ denote the average of the two quantities, i.e.,

$$L_{mid} = 0.5 * (L_{corr} + L_{incorr}) \qquad (7)$$

$L_{mid}$ acts as a boundary between the correct and incorrect samples, effectively creating two clusters whose centroids are given by $L_{corr}$ and $L_{incorr}$. Thus, samples with loss less than $L_{mid}$ in epoch E can be identified as Region 1 samples, as they are likely to be correct. Fig. 6 plots the efficacy of $L_{mid}$ across different epochs (fraction of correct inputs under $L_{mid}$). As desired, a majority of the correct samples ($> 95\%$) fall in

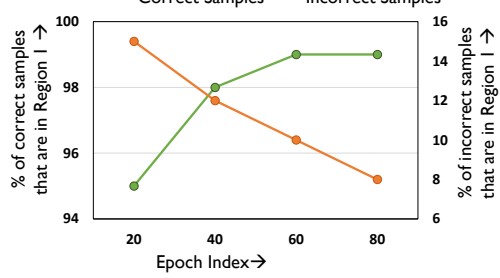

Figure 6: Efficacy of threshold $L_{mid}$

Region 1, while only including a negligible fraction of incorrect samples ($< 10\%$). Furthermore, samples with loss greater than $L_{incorr}$ in a particular epoch are in the upper percentile of the loss distribution of the incorrect samples. $L_{incorr}$ can hence used to create Region 2 and Region 3 as shown. We note that loss thresholds of better quality can potentially be identified using other techniques, such as by introducing hyper-parameters. However, tuning these hyper-parameters for each network separately is a costly process, diminishing the runtime benefits achieved by reducing training complexity.

We will now discuss the amenability criteria designed for samples belonging to Regions 1 and 3.

**Amenability Criteria for Region 1**: Consider a sample A belonging to Region 1 in epoch E, i.e., $Loss_A < L_{mid}$. From Figure 5(b) it is known that samples in Region 1 are likely to be correctly classified prior to mixing. We mix such samples as long as their loss does not exceed $L_{mid}$ at some later epoch $E'$, i.e., likely to be classified incorrectly. After epoch $E'$, they are shifted to $S_{noMix}$. Fig. 7 illustrates the temporal variation in the number of samples that are in $S_{mix}$, and from Region 1 of the loss distribution. As can be seen, the number of such samples increases across epochs. This is because as epochs progress classification accuracy improves, thereby resulting in more samples having loss below $L_{mid}$, i.e., belonging to Region 1. We note that using a loss-based threshold to determine amenability to mixing is more robust instead of directly using

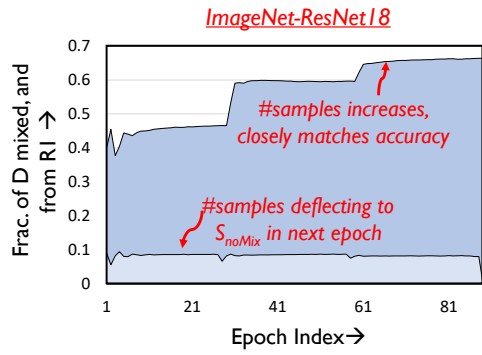

Figure 7: Amenability of Region 1

classification performance (Sec. 7), as we find that mixing outlier samples, i.e., samples with high loss yet correct classification affects overall accuracy.

The graph also depicts the fraction of samples that deflect to $S_{noMix}$ every epoch, which is a very small fraction of the samples that are mixed. This justifies the design of the amenability rule for Region 1.

**Amenability Criteria for Region 3**: Samples in Region 3 have high loss ($loss > L_{incorr}$), and are generally very difficult to classify by the network even if they are trained without mixing. In fact, we observe that a considerable fraction of samples that consistently occur in Region 3 across epochs remain incorrect at the end of the training process. Let $I$ denote the set of such samples that are incorrect when training concludes. We plot a histogram of the number of epochs samples in $I$ occupy Region 3 across training in Fig. 8(a). Clearly, it is observed that over half the samples in $I$ consistently occur in Region 3 for over 70%

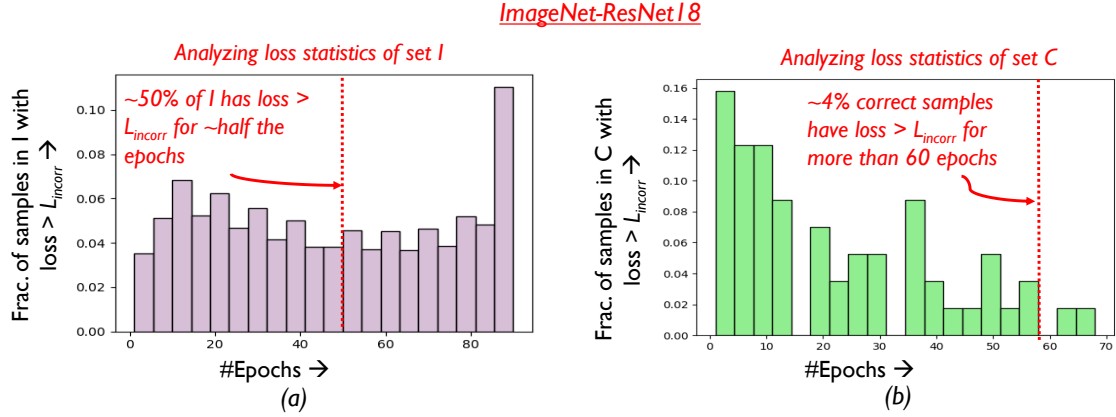

Figure 8: Loss dynamics of samples in set I and set C

of the training process. It can thus be argued from a practical runtime efficiency perspective that training effort on such samples can be reduced using mixing.

Some challenges however persist. As classification statistics evolve during training, it is difficult to determine which samples to mix at earlier epochs, without negatively affecting final classification accuracy. Consider set C, which comprises of samples that are correctly classified at the end of training. In Fig. 8(b), it is seen that around 4% of the samples in C occur in Region 3 for over 60% of the training process, with their classification accuracy improving only in the later stages of training. We must thus stipulate criteria to identify the desired subset of Region 3 samples that can be mixed.

To that end, we therefore target samples that the network finds difficult to classify at the moment, i.e., in the current epoch. In addition to belonging to Region 3, if a sample's loss increases over consecutive epochs (i.e., become increasingly difficult) it is mixed for the next epoch, ensuing which it is brought back to

$S_{noMix}$. In Fig. 9(b), we find that increasing the period of time $k$ for which the difficult samples must exhibit increasing loss and subsequently be mixed, only marginally improves the accuracy and runtime benefits. We hence use $k = 1$ for all our experiments thereby eliminating our dependence on any hyper-parameters. The temporal variation in the fraction of Region 3 samples mixed every epoch is depicted in Fig. 9(a). This fraction decreases across epochs, since several samples in Region 3 shift to Region 1 as accuracy improves. Interestingly, mixing difficult samples provides $\sim 0.2\%$ boost in classification performance over the overall validation set across all our benchmarks, as opposed to training them without mix-

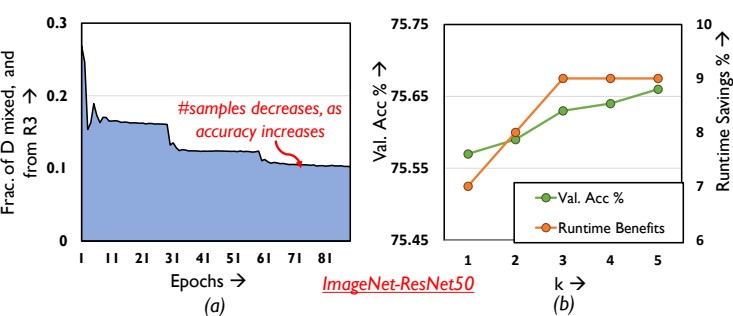

Figure 9: Amenability for samples in Region 3

ing. We believe this has the effect of allowing the network to focus on samples with moderate loss, that are more likely to contribute to final accuracy. Finally, we highlight the advantage of mixing such difficult samples instead of skipping them in Sec 4.

Determining sample amenability every epoch adds not more than 2% overhead in runtime on average, and 4% additional storage costs. The proposed amenability criteria thus help us successfully realize selective mixing, i.e., achieve a competitive runtime efficiency versus accuracy trade-off.

## 4 Experimental Results

We showcase the runtime benefits achieved by mixTrain across different classes of image recognition DNNs, namely convolutional neural networks (i.e., CNNs) and vision transformers Dosovitskiy et al. (2020). We consider two datasets, namely ImageNet (Deng et al., 2009) and Cifar10 (Krizhevsky et al.). The benchmarks for the ImageNet dataset consist of four image-recognition CNNs, *viz.* ResNet18, ResNet34, ResNet50 (He et al., 2015) and MobileNetV2 (Sandler et al., 2018), trained using the same training hyper-parameters such as learning rate, epochs etc., as in (He et al., 2015; Sandler et al., 2018). With regards to the Cifar10 dataset, we consider the ResNet18 and Resnet34 image-recognition CNNs (He et al., 2015). We also consider three vision transformer architectures, ViT-small, ViT-SWIN and ViT-pretrained. Details on the vision transformer architectures, and training hyper-parameters for all benchmarks can be found in Sec. 7.1.

Across all benchmarks, we report the speed-up achieved by mixTrain over the same number of epochs as the baseline, by comparing wall-clock times.

### 4.1 Execution Time Benefits

**ImageNet**: Table 1 presents the training performance of baseline SGD and mixTrain on different ImageNet benchmarks in terms of the Top-1 classification error and speed-up. On average, across all benchmarks, mixTrain mixes nearly 48% and 68% of the training dataset per epoch with MixUp and CutMix respectively. As can be seen, CutMix achieves a slightly superior trade-off than MixUp across all benchmarks, achieving upto around 1.6× reduction in runtime compared to to the baseline, while sacrificing only ∼0.2% loss in Top-1 accuracy. This is primarily because interference between constituent samples is better mitigated through split propagation, thereby resulting in more inputs being mixed.

**Cifar10**: We present our runtime and accuracy trade-off achieved on the Cifar10 vision transformer benchmarks in Table 2. As can be seen, mixTrain achieves 1.3×-1.6× training speed-up for nearly no loss in accuracy. This clearly underscores that mixTrain is directly applicable to any image classification DNN, regardless of the architecture or backbone deployed. Further, our results in Table 2 also indicate that mixTrain is not only applicable to training vision transformers from scratch, but to the fine-tuning stage as well. In Section 7.2 we discuss the speed-ups achieved by mixTrain on the CNN benchmarks trained on Cifar10.

*Runtime overhead analysis*: Across all our benchmarks, we observe that mixTrain adds no more than 2% overhead in runtime. These marginal overheads arise due to (i) calculating amenability of inputs to interpo-

Table 1: Training CNNs on ImageNet

| Network | Training Strategy | Top-1 Error | Speed-Up |
|---|---|---|---|
| ResNet18 | Baseline SGD | 30.2% | 1× |
| | mixTrain-CutMix | **30.44%** | **1.51×** |
| | mixTrain-MixUp | **30.6%** | **1.32×** |
| ResNet34 | Baseline SGD | 26% | 1× |
| | mixTrain-CutMix | **26.25%** | **1.54×** |
| | mixTrain-MixUp | **26.4%** | **1.37×** |
| ResNet50 | Baseline SGD | 24.3% | 1× |
| | mixTrain-CutMix | **24.45%** | **1.56×** |
| | mixTrain-MixUp | **24.6%** | **1.41×** |
| MobileNetV2 | Baseline SGD | 28.5% | 1× |
| | mixTrain-CutMix | **28.76%** | **1.52×** |
| | mixTrain-MixUp | **29%** | **1.3×** |

Table 2: Training vision transformers on Cifar10

| Network | Training Strategy | Top-1 Error | Speed-Up |
|---|---|---|---|
| ViT-small (Training from scratch) | Baseline SGD | 19% | 1× |
| | mixTrain-MixUp | **19.11%** | **1.37×** |
| | mixTrain-CutMix | **1.35%** | **1.32×** |
| ViT-SWIN (Training from scratch) | Baseline SGD | 9% | 1× |
| | mixTrain-MixUp | **8.9%** | **1.44×** |
| | mixTrain-CutMix | **9.2%** | **1.4×** |
| ViT-pretrained (Fine-tuning) | Baseline SGD | 2.5% | 1× |
| | mixTrain-MixUp | **2.46%** | **1.6×** |
| | mixTrain-CutMix | **2.55%** | **1.58×** |

lation and (ii) split propagation (for Cut-Mix). In (i) we compare the sample's loss against some thresholds, and update thresholds every epoch. However, these simple scalar operations have negligible runtime (<1.5% overhead) compared to the multiple GEMM operations performed during training. For (ii), during split propagation, the FC layers process the constituent inputs separately. However, the FC layers now operate on inputs of smaller size (i.e., corresponding to the size occupied by the features of the constituent input, which is nearly half the size of the original input). Thus, split propagation also adds less than <1% runtime overhead compared to the baseline.

## 4.2 Ablation Study

In this subsection we conduct an ablation analysis of mixTrain.

**Contribution of interference reduction and selective mixing**: mixTrain uses two strategies to achieve an optimal accuracy versus runtime trade-off, i.e., reducing impact of interference and selective mixing. Fig. 10(a) depicts the contribution of each strategy towards runtime savings, for the CutMix operator. The light blue markings indicate naive mixing. Selective mixing automatically identifies a subset of training samples that can be mixed every epoch such that classification accuracy is not impacted. However, if interference between the constituent inputs is not

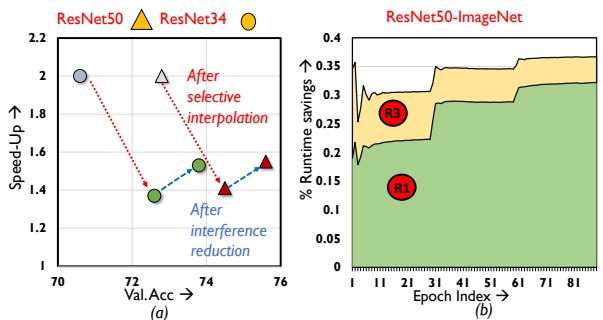

Figure 10: Ablation analysis

mitigated, training performance on mixed samples is poor (green markings). Consequently, the selective mixing strategy is forced to become conservative, identifying fewer samples that can be mixed every epoch

without affecting accuracy severely. Reducing interference between the constituent inputs improves accuracy by more than 1%, and speed-up by 10% (red markings).

**Breakdown of selective interpolation**: We breakdown selective interpolation by examining the region of the loss distribution that provides the most benefits. From Fig. 10(b) (generated using `CutMix`) it is evident that Region 1 samples provide the bulk of our benefits on the ResNet18-ImageNet benchmark, accounting for nearly 25% of the savings. This is because as training progresses, a majority of training samples fall in Region 1. Interpolating Region 3 samples, accounts for additional 8% runtime savings.

### 4.3 Quantitative Comparison Study

We compare the performance of mixTrain against competing methods that accelerate DNN training.

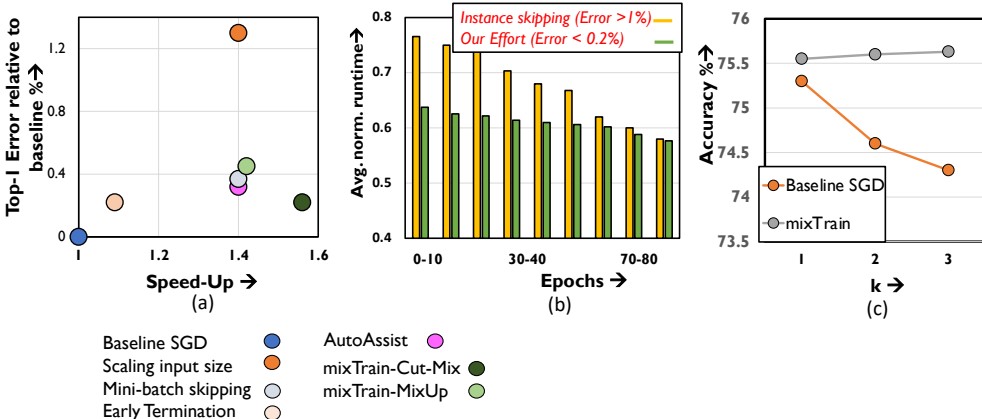

Figure 11: Quantitative Comparison Study

**Instance Skipping**: As a representative of instance skipping, we specifically consider the performance of (Zhang et al., 2019) (Fig. 11(a)) and (Jiang et al., 2019) (Fig. 11(b)) on the ResNet50 benchmark. In these techniques samples that the network finds easy to classify, as identified by low classification loss, are skipped thereby resulting in fewer mini-batches as training proceeds. Two issues are typically encountered by such techniques. First, as no training is conducted on the samples that are skipped, this subset is often a small, conservative fraction of the training dataset. Second, additional overhead is incurred in each epoch to determine this subset, as it is non-trivial to estimate the most recent loss of samples that had been discarded in previous epochs. In Fig. 11(b), we implement (Jiang et al., 2019) and overlook the overheads associated in determining the subset of samples that must be skipped, and report the resulting runtime across epochs.

Clearly, mixTrain achieves better model accuracy and runtime benefits against both efforts, even when overheads are overlooked. As the network is ultimately trained on every input in each epoch, we reduce the number of minibatches more aggressively, while incurring negligible overheads incurred to form $S_{mix}$ and $S_{noMix}$. Finally, we analyze the accuracy if Region 3 samples were to be skipped instead of mixed, using the same policy discussed in Sec. 3.2 for different values of $k$. Clearly, mixTrain achieves better convergence, allowing it to leverage runtime benefits from this region.

**Coreset selection techniques**: In the table below, we compare the performance of MixTrain-CutMix against three popular coreset selection techniques: Glister (Killamsetty et al., 2020), Grand (Paul et al., 2021) and Facility-location based methods (Iyer et al., 2020). Similar to mixTrain, coreset selection techniques aim to reduce training runtime by reducing the number of mini-batches to train every epoch, by identifying a subset of training data-points that are critical to accuracy. Such techniques perform better than random sampling (i.e., better accuracy), when the fraction of the training dataset retained is low (Guo et al., 2022). However, as can be seen in Table 3, these techniques require a large fraction of the training dataset in order to remain iso-accurate with the baseline. mixTrain clearly achieves a better accuracy versus speed-up trade-off.

**Other approximations**: We consider three approximation strategies, i.e., early termination, mini-batch skipping and input size scaling (Fig.Fig. 11(a)). For early-termination, we stop baseline SGD training at

Table 3: Comparison against coreset selection techniques

| Training Method | Average fraction of the dataset used for training across epochs | Top-1 Error | Speed-Up |
|---|---|---|---|
| Baseline | 1 | 4.4% | 1× |
| `mixTrain-MixUp` | **0.69** | **4.33%** | **1.4×** |
| `mixTrain-CutMix` | **0.66** | **4.2%** | **1.45×** |
| Glister | 0.8 | 4.65 | 1.18× |
| | 0.7 | 4.76% | 1.32× |
| Grand | 0.8 | 4.6% | 1.15× |
| | 0.7 | 4.7% | 1.2× |
| Facility Location | 0.8 | 4.55% | 1.19× |
| | 0.7 | 4.79% | 1.25× |

an earlier epoch when it achieves the same accuracy as mixTrain, and report the resulting runtime benefits. Next, for mini-batch skipping we stochastically skip $s\%$ of the mini-batches every epoch, and for input size scaling, we train on inputs scaled down by some factor $s$. In both cases, the parameter $s$ is selected such that it is iso-runtime with mixTrain. Clearly, in all three cases, mixTrain achieves a superior accuracy versus runtime trade-off as seen for the ResNet50 benchmark.

## 5 Related Work

We now discuss related research efforts to accelerate DNN training.

**Hyper-parameter tuning**: Many notable efforts are directed towards achieving training efficiency by controlling the hyper-parameters involved in gradient-descent, notably the learning rate and momentum. (You et al., 2017; Akiba et al., 2017; Goyal et al., 2017) propose learning rate tuning algorithms that achieve training in less than an hour with no loss in accuracy, when distributed to over hundreds of CPU/GPU cores.

**Optimizers with fast convergence**: This class of efforts includes optimizers that achieve improved generalization performance within a certain training budget. These techniques target the evaluation of the weight gradient every iteration- for example, optimizers such as AvaGrad (Savarese et al., 2019) and Adam (Kingma & Ba, 2015) adaptively compute the learning rate across training epochs, resulting in faster convergence than SGD in a similar number of epochs for certain tasks. Similarly, techniques such as (Sutskever et al., 2013) utilize a momentum parameter during training to achieve faster convergence.

**Model size reduction during training**: Model size reduction investigates dynamically pruning (Yuan et al., 2020) or quantizing (Sun et al., 2019) a model during training itself. Training a reduced-capacity model, or with lower-precision results in training speed-ups.

**Coreset selection strategies**: Such techniques select a subset of the training samples that are most informative, i.e., critical to accuracy. These techniques differ in the identification of such critical training samples. Commonly used methods to determine a sample's importance include analyzing sample loss (Jiang et al., 2019; Zhang et al., 2019), gradient-matching techniques (Killamsetty et al., 2021), bi-level optimization methods (Killamsetty et al., 2020), sub-modularity based approaches (Iyer et al., 2020), and decision boundary based methods (Margatina et al., 2021).

## 6 Conclusion

We introduce a new approach to improve the training efficiency of state-of-the-art DNNs by utilizing input mixing. We propose mixTrain that comprises of two strategies to achieve an acceptable accuracy versus speed-up trade-off. First, we propose split propagation and adaptive mixing to reduce the impact of interference between the constituent inputs in a composite sample. Second, we apply mixing selectively, i.e., only on a subset of the training dataset every epoch. Across DNNs on the ImageNet dataset, we achieve upto a 1.6× improvement in runtime for ~0.2% loss in accuracy.

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

# 7 Appendix

## 7.1 Experimental Setup

This subsection describes the experimental setup used for realizing the baseline and proposed training schemes, on the benchmarks specified in Section 4 of the main paper. We conduct our experiments on the complete training and test datasets of each benchmark, using the PyTorch (Paszke et al., 2019) framework.

**Baseline**: We consider SGD training as the baseline in our experiments. The hyper-parameters used in SGD training of each of the benchmarks are described below.

ImageNet: For experiments in Section 4.1 we utilize a batch-size of 64 per GPU, for all benchmarks. For the ResNet18, ResNet34 and ResNet50 benchmarks the initial learning rate set to 0.025. The learning rate is decreased by 0.1 every 30 epochs, for a total training duration of 90 epochs, and the weight decay is $4e-5$. The MobileNetV2 benchmark utilizes an initial learning rate of 0.0125. We use a cosine learning rate decay schedule, as in (Li et al., 2019) for 150 epochs. The weight decay is set to $4e-5$. All benchmarks use an input size of 224*224*3.

Cifar10-CNNs: All experiments on the convolutional neural networks (i.e., ResNet18 and ResNet34) utilize a batch-size of 128, trained on a single GPU. We consider two different hyper-parameter settings that differ in the learning rate schedule used. When using a linear learning rate schedule, all benchmarks are trained with an initial learning rate of 0.05 that is decayed by 0.1 every 10 epochs, across 90 epochs. The cosine annealing learning rate schedule uses an initial learning rate of 0.1, that is gradually decayed over 200 epochs. Across all experiments, the weight decay is set to $5e-4$. All benchmarks utilize an input size of 32*32*3.

Cifar10-Transformers: We consider three vision transformer architectures, ViT-small, ViT-SWIN and ViT-pretrained. The ViT-small architecture has a patch-size of (4*4), with the hidden dimension size equal to 512. The network consists of 8 attention heads, and a depth of 6. The ViT-SWIN architecture is identical to the Swin-T architecture in (Liu et al., 2021). When training from scratch, both networks operate on inputs of size (32*32*3), and are trained for 100 epochs using a cosine annealing learning rate schedule, with the initial learning rate of 1e-4. For the fine-tuning experiment, the ViT-pretrained network uses the ViT-B/16 architecture described in (Dosovitskiy et al., 2020). Here the pretrained weights are obtained by training on the ImageNet-21k dataset (Deng et al., 2009), and the network hence accepts an input of size (384*384*3). Fine-tuning is conducted for 3 epochs. For all models, training is conducted across 4 GPUs, with the batch-size set to 128.

mixTrain: mixTrain uses the same learning rate, weight decay, and number of epochs as baseline SGD, requiring no additional hyper-parameters. We use the same random seed for both our baseline and mix-Train experiments. Results in Sec. 4.1 are reported by averaging over 3 different training runs.

Table 4: Training CNNs on Cifar10 using mixTrain

| Network | Training Strategy | Top-1 Error | Speed-Up |
|---|---|---|---|
| ResNet18 | Baseline SGD (linear learning rate schedule) | 6.5% | 1× |
|  | mixTrain-CutMix | **5.4%** | **1.74×** |
|  | mixTrain-MixUp | **5.7%** | **1.69×** |
| ResNet18 | Baseline SGD (cosine annealing learning rate schedule) | 4.4% | 1× |
|  | mixTrain-CutMix | **4.2%** | **1.45×** |
|  | mixTrain-MixUp | **4.33%** | **1.41×** |
| ResNet34 | Baseline SGD (linear learning rate schedule) | 5.2% | 1× |
|  | mixTrain-CutMix | **4.2%** | **1.78×** |
|  | mixTrain-MixUp | **4.6%** | **1.71×** |

Table 5: mixTrain with different optimizers

| Network | Training Strategy | Top-1 Error | Speed-Up |
|---|---|---|---|
| ResNet18 | Baseline Adam | 6% | 1× |
|  | mixTrain-CutMix | **5.8%** | **1.59×** |
|  | mixTrain-MixUp | **5.92%** | **1.51×** |
| ResNet18 | Baseline AvaGrad | 5.7% | 1× |
|  | mixTrain-CutMix | **5.2%** | **1.42×** |
|  | mixTrain-MixUp | **5.4%** | **1.4×** |
| ResNet18 | Baseline AvaGrad-W | 5.8% | 1× |
|  | mixTrain-CutMix | **5.28%** | **1.44×** |
|  | mixTrain-MixUp | **5.3%** | **1.39×** |

## 7.2 Experimental Results on Cifar10

To underscore the wide applicability of mixTrain, we present our runtime and accuracy trade-off achieved on the Cifar10 benchmarks in Table 4. Across our benchmarks, MixUp achieves upto 1.7 × improvement in runtime, while CutMix achieves a 1.8× runtime improvement. Clearly, both mixing strategies provide a boost in accuracy, due to the improved regularization provided via mixing samples.

As can be seen in Table 5, we also highlight the applicability of mixTrain to optimizers such as Adam (Kingma & Ba, 2017) and AvaGrad (Savarese et al., 2019), and different learning rate schedules (Loshchilov & Hutter, 2016). Typically, such optimizers propose techniques to evaluate the weight gradients in a manner that results in faster convergence. MixTrain does not interfere with the evaluation of such weight gradients-regardless of the optimizer used, MixTrain achieves training acceleration by reducing the effective size of the dataset to iterate over each epoch. As can be seen, MixTrain can be successfully applied in conjunction with such optimizers.

## 7.3 Analysis of Top-5 accuracy without interference reduction

In Sec. 3.1 we mention that the network appears to be unable to detect both constituent inputs when interference is not reduced. At most, the network detects only one of the constituent inputs, with the second constituent rarely appearing in the Top-5 predictions made. We provide the Top-5 classification accuracy of the second constituent, prior to reducing interference.

This necessitates the need for devising strategies to reduce interference between the constituent inputs of a composite sample.

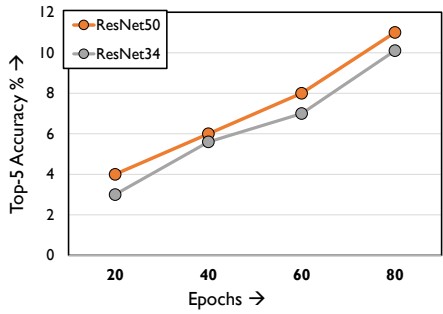

Figure 12: Top-5 classification accuracy of second constituent input

## 7.4   Training runtime

We present our training runtime results in Table 6. Note that the Cifar10 experiments are conducted on a single Nvidia RTX 2080Ti machine, while the ImageNet experiments are conducted across 4 RTX gpus.

Table 6: Training runtime

| Network | Training Strategy | Runtime |
|---|---|---|
| | Baseline-SGD | 3.6 hours |
| ResNet18-Cifar10 | mixTrain-**CutMix** | 1.95 hours |
| | Baseline-SGD | 30 hours |
| ResNet18-ImageNet | mixTrain-**CutMix** | 20.1 hours |
| | Baseline-SGD | 51 hours |
| ResNet50-ImageNet | mixTrain-**CutMix** | 32.7 hours |

## 7.5   Analyzing mixTrain **when mixing more than 2 inputs**

mixTrain can be extended to beyond 2 samples. However, we observe that the effect of mixing more than two samples is different for different benchmarks. Table 7 below shows the performance of mixTrain when 2 and 3 samples are mixed using the Cut-Mix operator, on the Cifar10 and ImageNet datasets for the ResNet18 network. For the Cifar10-ResNet18 benchmark, mixing N=3 samples clearly provides better runtime savings than N=2, and at comparable accuracy to baseline. However, we observe a noticeable drop in accuracy for the ImageNet benchmark. In the context of Cut-Mix, this is because the class object of interest occupies a smaller fraction of the input area for ImageNet, and is likely to be missed in the random Cut-Mix patch. We note that we observe similar trends in accuracy when using the Mix-Up operator. Here, the interference between constituent inputs is higher due to averaging pixel information across more samples.

## 7.6   Analysis of loss across consecutive epochs

As mentioned in Sec. 3.2, we utilize the loss of a sample in epoch E to determine it's amenability to mixing in epoch E+1. However, several mini-batches pass before a sample is trained again in the next epoch. As the model undergoes many changes to its weights, it is possible that the loss of a sample in epoch E might be quite substantially different from that in epoch E+1.

Fig. 13 plots the loss curve averaged across all training examples when trained with SGD. The loss appears to change rapidly only for the first few epochs, and later when the learning rate changes. In other periods, changes in loss happen more gradually. We find that the same analysis is generally applicable when samples are mixed as well. This thus justifies using the loss in epoch E to justify amenability in epoch E+1.

Table 7: Mixing more than 2 inputs using mixTrain

| Network | Training Strategy | Top-1 Acc | Speed-Up |
|---|---|---|---|
| ResNet18-Cifar10 | Baseline SGD (N=0) | 95.6% | 1× |
| | mixTrain-**CutMix** (N=2) | **95.8%** | **1.45×** |
| | mixTrain-**CutMix** (N=3) | **94.4%** | **2.32×** |
| ResNet18-ImageNet | Baseline SGD (N=0) | 69.8% | 1× |
| | mixTrain-**CutMix** (N=2) | **69.56%** | **1.5×** |
| | mixTrain-**CutMix** (N=3) | **68.7%** | **1.9×** |

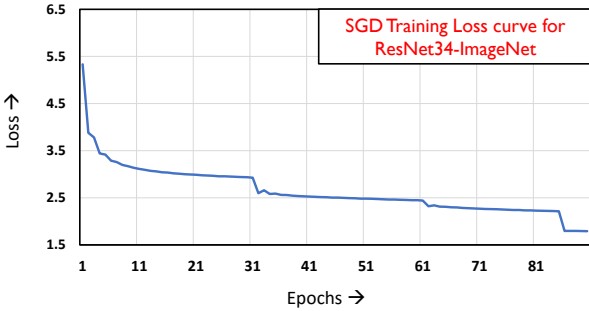

Figure 13: Change in average loss across epochs

## 7.7 Analyzing efficacy of amenability metric

As part of our key strategies to achieve a good accuracy versus runtime efficiency trade-off, we propose selectively mixing samples in Section 3.2. We take into consideration the region of the loss distribution where the sample occurs in epoch E to appropriately decide whether the sample should be mixed in the next epoch. Each region differs based on the estimated impact mixing a sample may have on accuracy. Consequently, each region has its own criteria for gauging amenability for the next epoch.

In Table 8, we compare our proposed selective mixing strategy against the following set of rules to gauge amenability.

Table 8: Analyzing efficacy of our amenability metric

| Benchmark | Amenability Metric | Top-1 err | Speed-Up |
|---|---|---|---|
| ResNet50 | Our Effort (Region 1 only) | 24.56% | 1.38 |
| | Our Effort (Regions 1 and 3) | 24.45% | 1.56 |
| | Threshold = Accuracy | 24.80% | 1.36 |
| | Threshold = Average Loss | 24.50% | 1.33 |
| | Region 1 and threshold = $L_{incorr}$ for Region 3 | 25.14% | 1.74 |

- First, we analyze the trade-off achieved when we mix inputs that were correctly classified in previous epoch, instead of using $L_{mid}$ as in Sec. 3.2. Essentially, only those samples that are correct in epoch E are mixed in the next epoch. We find that the $L_{mid}$ threshold (Row 1) achieves slightly better classification accuracy, as outlier inputs with correct classification are avoided.

- Next, we compare against an average loss threshold, i.e., we calculate the running average of the loss across all the samples in $S_{noMix}$. If a sample in epoch E has loss lower than the running average, it is mixed in the next epoch and vice-versa. As can be seen, our $L_{mid}$ threshold (Row 1) achieves better speed-ups for nearly the same classification accuracy. Across epochs, classification accuracy improves and average loss reduces, often with several correctly classified samples with loss above the average loss. This metric thus loses the opportunity to approximate training effort on these these correctly classified samples that are amenable to interpolation.

- Finally, we compare the efficacy of our Region 3 criterion. We observe the trade-off achieved when all samples above $L_{incorr}$ are mixed, in addition to Region 1 approximations. Clearly, our proposed criterion attains better accuracy (Row 2).

## 7.8 Discussion on applicability of mixTrain to other tasks

We now discuss the applicability of mixTrain to different kinds of tasks other than image classification, such as natural language processing and semantic segmentation.

In image classification problems, mixing inputs does not hinder the DNN's ability to recognize the features of the constituent inputs [1,2], as the DNN's input space is continuous. Thus, the mixed input can be considered to be a new data point in the input space. However, on tasks such as NLP, each word in a sentence is from a discrete dictionary, represented using a token generated using an embedding table. Thus, the new token generated by, for example, linearly averaging the tokens of $N$ constituent sentences, may not exist in the dictionary the network is trained on. Furthermore, networks designed for such tasks factor into consideration the position of each word in a sentence, and their relation to other words. The impact of mixing tokens of different sentences is thus unclear. Consequently, MixTrain may not be applicable to such tasks.

Moving on to semantic segmentation tasks, each pixel in the input image is assigned a target label that indicates the class to which the pixel belongs to. In the context of the MixUp operator, after mixing has been applied, each pixel belongs to two classes, pertaining to each of the constituent inputs. The final target label of the pixel in the composite sample can thus be obtained by averaging the one-hot encoded labels of the corresponding pixel in the constituent inputs. In the context of the CutMix operator, the composite sample contains patches from each of the constituent inputs. Thus, the labels of pixels in a patch of the composite sample can be easily inferred by their value in the original constituent input. However, while such CutMix and MixUp based training for segmentation tasks have been demonstrated in a semi-supervised context, their usage in fully supervised settings has not been widely adopted. Thus, extending MixTrain to such tasks is unclear, and beyond the scope of this work.

