# OpenReview forum: "MixTrain: Accelerating DNN Training via Input Mixing"
_TMLR — Rejected by TMLR_

### Review · Reviewer_CX76 · 2023-05-11

**Summary Of Contributions:**

This paper proposes a novel training method called MixTrain to address the issue of accelerating training deep models. MixTrain proposes that the main cause of the poor performance of naïve mixing is the interference between constituent inputs. Thus , split propagation and adaptive mixing is introduced for the problem. Selective mixing is proposed to improve the accuracy.

**Audience:**

Yes

**Claims And Evidence:**

Yes

**Requested Changes:**

See Weaknesses

**Strengths And Weaknesses:**

Strengths:
The paper is generally well written, clearly structured and quite easy to follow.
The empirical results seem good on most of the performed tasks.

Weaknesses:
1.	Too many typos and incorrect format. For example
1)We show that mixTrainachieves superior accuracy vs. efficiency tradeoffs
2)Naturally, a simple boost in performance can be achieved by atleast improving the loss for one of the constituent inputs of the mixed input.

Besides, the abstract should not be divided into two paragraphs.

2.	DNN does not only include CNN. It seems that the method can be used in some SOTA backbones, such as deit, swin transformer, etc. Some experiments should be conducted.

3.	This method should be compared with the other DNN Training acceleration methods.

---

> ### Author Response · Authors · 2023-06-13
> **Response to Reviewer CX76**
>
> Re: Correction in typos
>
> We apologize for the typographical errors, and have updated the manuscript, correcting the same.
>
> ==============================================================
>
> Re: Applicability to other architectures:
>
> The reviewer is absolutely correct in pointing out that MixTrain is applicable to other image classification DNNs such as Vision Transformers.  To demonstrate the same, we have conducted experiments wherein we apply MixTrain when training Vision Transformers [1,2] on the Cifar10 dataset. Our new results, reported in the revised paper (Table 2, Sec. 4.1), suggest that the proposed approach achieves considerable speed-up while maintaining accuracy. We have also provided details on the architectures and training hyper-parameters in Sec. 7.1 of the updated paper.
>
> [1]: An Image is Worth 16x16 Words: Transformers for Image Recognition at Scale. ICLR 2021
>
> [2]: Swin Transformer: Hierarchical Vision Transformer using Shifted Windows. ICCV 2021
>
> =========================================================
>
> Re: Comparisons against other training acceleration techniques
>
> We thank the reviewer for this suggestion. In Table 3 (Sec. 4.3) of the revised paper, we compare the performance of MixTrain-CutMix against three popular coreset selection techniques: Glister [1], Grand [2] and Facility-location based methods [3]. Similar to MixTrain, coreset selection techniques aim to reduce training runtime by reducing the number of mini-batches to train every epoch, by identifying a subset of training data-points that are critical to accuracy. Such techniques perform better than random sampling (i.e., better accuracy), when the fraction of the training dataset retained is low [4]. However, as can be seen in Table 3, these techniques require a large fraction of the training dataset in order to remain iso-accurate with the baseline. MixTrain clearly achieves a better accuracy versus speed-up trade-off.
>
> [1]: Glister: Generalization based data subset selection for efficient and robust learning. AAAI 2020
>
> [2]: GraNd: Deep learning on a data diet: Finding important examples early in training. NeurIPS 2021
>
> [3]: Submodular combinatorial information measures with applications in machine learning. PMLR 2021
>
> [4]: DeepCore: A Comprehensive Library for Coreset Selection in Deep Learning. ArXiv 2022

---

### Review · Reviewer_aeAv · 2023-05-20

**Summary Of Contributions:**

This paper introduces MixTrain, a new approach for accelerating the training speed of DNN models. The key idea of MixTrain is to mix two inputs into one. By doing so, one may reduce the number of samples in one epoch for training and lead to potential training time reduction. The challenge is how to make sure this scheme would lead to good accuracy. To tackle this challenge, the paper explores two input mixing schemes: CutMix and Mixup. The former uses a random cut to an image and replaces a part of another image with this cut, while the latter uses a weighted average of two inputs to generate the mixed input. Naively applying the above schemes would lead to an accuracy drop. Therefore, the authors propose several fixes to mitigate the accuracy drop: (1) split propagation, which sequentially processes unmixed inputs to avoid interference, (2) adaptive mixing, which adaptively changes the weights when doing MixUp, and (3) selective mixing, maintain a subset of samples for input mixing and dynamically change that subset based on loss dynamics during training. With those optimizations, the paper shows that MixTrain achieves up to 1.6x training speedup on ResNets over ImageNet/CIFAR with 0.2 accuracy drop.

**Audience:**

Yes

**Broader Impact Concerns:**

None.

**Claims And Evidence:**

Yes

**Requested Changes:**

Changes that are critical for a recommendation for acceptance:

- Comparison with SOTA approach such as https://openreview.net/pdf?id=493VFz-ZvDD.

- In adaptive mixing, the losses of samples from the previous epochs are needed to calculate the weights. However, the inputs are mixed in the previous epoch, e.g., one loss for mixed input 1&2, so how is it possible to get the separate loss of input 1 and 2 from the previous epoch without incurring additional cost? A runtime overhead discussion is needed.

Items that would strengthen the work:

- Split propagation splits mixed input and processes them separately through fully connected layers (FCL). This seems to reduce the compute granularity of FCL and slows down the training. For models with many FCLs, the approach's speedup would be further limited. A clarification is needed.

- While split propagation may be applicable to CutMix to reduce interference, where the split is clear, it does not seem to be applicable to Mixup, where all elements in an intermediate feature map have a dependency on both inputs. Some clarification is needed.

- No real training time has been reported in the paper. It would be better to report the actual training time to cross-reference the results.

- Would the proposed method be extended to mixing up more than two inputs? It would be good to include a study of how mixing multiple inputs (>2) would affect the accuracy-time tradeoff.

- In the last paragraph of Section 3.1, "by atleast improving" -> "by at least improving".

**Strengths And Weaknesses:**

Strengths:

- Interesting idea of selectively using mixed inputs to accelerate training speed.
- The amenability evaluation for selective mixing provides some good insight on how to get input with and without mixing loss values in a lightweight way.
- The approach seems to lead to promising training speedup without incurring additional hyperparameter tuning on tested datasets.

Weaknesses:

- Training ResNets on ImageNet does not pose major training cost challenges. Training ResNet-50 on ImageNet takes no more than 2 days on lower-end GPUs and within less than 15 mins in more recent advanced hardware: https://developer.nvidia.com/deep-learning-performance-training-inference. While the authors may argue the approach is applicable to other tasks, there is no evidence to support so at this point.

 - The paper compares MixTrain with several baseline methods and shows a better accuracy-time tradeoff. However, those baselines appear to be weak. For instance, https://openreview.net/pdf?id=493VFz-ZvDD reduces training cost via a combination of data sieving and other techniques.  It would be better if the paper can make a comparison with more recent and stronger baselines.

---

> ### Author Response · Authors · 2023-06-13
> **Response to Reviewer aeAv - Part 1**
>
> We thank the reviewer for their insightful feedback, and address their concerns below
>
> Re: Comparison with SOTA approach
>
> We thank the reviewer for this suggestion. The paper suggested by the Reviewer (https://openreview.net/pdf?id=493VFz-ZvDD) reports training FLOPs reduction by (a) exploiting fine-grained sparsity in the weights of the DNN during training (b) data-sieving techniques to reduce the size of the training dataset. However, it can be inferred from Appendix E in the paper that exploiting sparsity contributes to a major fraction of their reported training FLOPS reduction (>85%). Moreover, not all commodity hardware platforms (CPUs/GPUs) can extract speed-ups from fine-grained sparsity, due to the inherent irregularity in the computations. While this previous work reports reductions in FLOPs, the training time improvements on CPUs/GPUs will be diminished. In contrast, MixTrain achieves training runtime speed-ups (1.6x-1.8x) by reducing the size of the dataset every epoch, i.e., fewer mini-batches to train every epoch, and does not interfere with the internal computations of the DNN. It can hence be applied across a wide range of platforms.
>
> In Table 3 (Sec. 4.3) of the revised paper, we compare the performance of MixTrain-CutMix against three popular coreset selection techniques: Glister [1], Grand [2] and Facility-location based methods [3]. Similar to MixTrain, coreset selection techniques aim to reduce training runtime by reducing the number of mini-batches to train every epoch, by identifying a subset of training data-points that are critical to accuracy. Such techniques perform better than random sampling (i.e., better accuracy), when the fraction of the training dataset retained is low [4]. However, as can be seen in Table 3, these techniques require a large fraction of the training dataset in order to remain iso-accurate with the baseline. MixTrain clearly achieves a better accuracy versus speed-up trade-off.
>
> [1]: Glister: Generalization based data subset selection for efficient and robust learning. AAAI 2020
>
> [2]: GraNd: Deep learning on a data diet: Finding important examples early in training. NeurIPS 2021
>
> [3]: Submodular combinatorial information measures with applications in machine learning. PMLR 2021
>
> [4]: DeepCore: A Comprehensive Library for Coreset Selection in Deep Learning. ArXiv 2022
>
> ====================================================================================
>
> Re: Loss of samples during adaptive mixing and runtime overhead discussion
>
> We elaborate the evaluation of the loss of a sample during mixing as follows. Consider a sample X that has been formed by mixing inputs X1 and X2 in some ratio alpha, at epoch E. Further, let the target labels of X1 and X2 be denoted as y1 and y2 respectively. At the end of epoch E, the loss of X is calculated as shown below in Equation 1. Note that we use cross-entropy loss.
>
> 	Loss(X) = alpha*Loss(X, y1) + (1-alpha)*Loss(X, y2)		Eq. 1
>
> We obtain an approximation for the loss of the individual inputs X1 and X2, shown in Equations 2 and 3. Here, we use the loss of the network on the mixed input to estimate its loss on the individual constituent inputs.
>
>  	 Loss(X1) = Loss(X,y1)								Eq. 2
>
> 	 Loss(X2) = Loss(X, y2)								Eq. 3
>
> We note that estimating the loss of X1 and X2 in such a manner is indeed an approximation- however, this allows us to avoid an additional forward propagation step to estimate the true loss of X1 and X2, thereby alleviating any runtime overhead. Moreover, we find that such an approximation does not affect training convergence, as observed in Sec. 4.1 of the paper. We have updated the paper to include this discussion in Sec. 3.1 (page 5).
>
> ==========================================================================
>
> Re: Runtime overhead discussion
>
> Runtime overheads arise due to (i) calculating amenability of inputs to interpolation (ii) split propagation (for Cut-Mix).
> In (i) we compare the sample’s loss against some thresholds, and update thresholds every epoch. However, these simple scalar operations have negligible runtime (~1.5% overhead) compared to the multiple GEMM operations performed during training.
> For (ii), during split propagation, the FC layers process the constituent inputs separately. However, the FC layers now operate on inputs of smaller size (i.e., corresponding to the size occupied by the features of the constituent input, which is nearly half the size of the original input). Thus, split propagation also adds less than <1% runtime overhead compared to the baseline. The paper has been updated to include this discussion, in Sec. 4.1

---

> > ### Author Response · Authors · 2023-06-13
> > **Response to Reviewer aeAv - Part 2**
> >
> > Re: Applicability of split propagation to Mix-Up
> >
> > Split Propagation and Adaptive Mixing are strategies that we introduce to improve the quality of the training on mixed inputs. When CutMix is applied, split propagation exploits the spatial separation of the features of the constituent inputs in a mixed sample. The reviewer is correct in pointing out that split propagation cannot be used with Mix-Up, as it is non-trivial to separate the features of the constituent inputs once they have been averaged. In this case, we only use Adaptive mixing, wherein the pixels of the constituent inputs are averaged, with more weightage placed on the more difficult constituent input. As can be seen in Sec. 4, we find that applying Adaptive Mixing to Mix-up based training is sufficient to achieve a good accuracy versus speed-up trade-off.
> >
> > ===============================================================
> >
> > Re: Training runtime
> >
> > We thank the reviewer for this suggestion. We present our training runtime analysis in Table 5 (Sec. 7.4) of the revised paper. Note that for the Cifar10 experiments listed, training is conducted on a single Nvidia RTX 2080Ti machine, while the ImageNet experiments are conducted across 4 RTX GPUs.
> >
> > ==================================================================
> >
> > Re: Applicability of MixTrain to more than 2 inputs
> >
> > In principle, mixTrain can be extended to beyond 2 samples. However, we observe that the effect of mixing more than two samples is different for different benchmarks. Table 6 (Sec. 7.5) of the revised paper shows the performance of mixTrain when 2 and 3 samples are mixed using the Cut-Mix operator, on the Cifar10 and ImageNet datasets for the ResNet18 network.
> >
> > For the Cifar10 benchmark, mixing N=3 samples clearly provides better runtime savings than N=2, and at comparable accuracy to baseline. However, we observe a noticeable drop in accuracy for the ImageNet-ResNet18 benchmark. In the context of Cut-Mix, this is because the class object of interest occupies a smaller fraction of the input area for ImageNet, and is likely to be missed in the random Cut-Mix patch. We note that we observe similar trends in accuracy when using the Mix-Up operator. Here, the interference between constituent inputs is higher due to averaging pixel information across more samples.
> >
> > We hope to expand and develop a mixing strategy that allows for mixing more than 2 inputs, as part of future experiments.

---

### Review · Reviewer_n2ZZ · 2023-05-31

**Summary Of Contributions:**

This work investigates the feasibility of leveraging mixup to accelerate DNN training. The motives of this work are quite simple -- train the model on the mixed inputs/labels to reduce the cost of training. This work finds the performance drop of simple linear combination and proposes the advanced approach to address the issue. Authors use several CNN architectures together with CIFAR-10 and ImageNet to evaluate their proposals. Significant speedup could be observed with marginal performance drop.

**Audience:**

Yes

**Broader Impact Concerns:**

Not applicable here.

**Claims And Evidence:**

Yes

**Requested Changes:**

Please address W1 and W2 above.

**Strengths And Weaknesses:**

S1. this work studied an interesting problem, while the motivation of adopting mixup for training acceleration was quite straightforward.
S2. authors found the limitation of the vanilla mixup for training acceleration, and proposed advanced approach.

W1. All experiments are based on convolutional architectures, such as MobileNet and ResNet-50, while the sizes of neural nets are quite small. These nets might no longer be considered as SOTA of image classification. Please include backbones such as ViT and incorporate more optimizers into the consideration/comparisons. It is reasonable to assume the proposed method would cause significant performance drop with modern architectures, as they perform better.

W2. This work studied the problem of image classification, it would be good to discuss the scenarios, such as natural language processing or semantic segmentation. In these cases, how can we perform mixup to accelerate training procedures, or whehter it is possible to mix images as the input and segmentation masks as the label.

---

> ### Author Response · Authors · 2023-06-13
> **Response to Reviewer n2ZZ**
>
> Re: Experiments on different backbones:
>
> We thank the reviewer for this helpful suggestion. MixTrain reduces the size of the training dataset dynamically, by analyzing the loss of a network on a particular input- in other words, MixTrain is directly applicable to any image classification network regardless of the architecture or backbone, with no additional changes compared to CNNs. To demonstrate this, we have conducted experiments wherein we apply MixTrain when training Vision Transformers [1,2] on the Cifar10 dataset. The results, reported in the revised paper (Table 2, Sec. 4.1), indicate that MixTrain achieves considerable speed-up while maintaining accuracy. We have also provided details on the architectures and training hyper-parameters in Sec. 7.1 of the updated paper.
>
> We would like to point out that SOTA networks are in fact more likely to benefit from MixTrain, as the fraction of training samples such networks find ‘easy’ to classify are higher- hence, more samples are mixed. As can be seen in Table 2 of the paper, as the network accuracy increases the speed-up achieved by MixTrain improves, with no impact on accuracy.
>
> [1]: An Image is Worth 16x16 Words: Transformers for Image Recognition at Scale. ICLR 2021
>
> [2]: Swin Transformer: Hierarchical Vision Transformer using Shifted Windows. ICCV 2021
>
> =============================================
>
> Re: Applicability to other optimizers
>
> To demonstrate the applicability of our approach to other optimizers, we apply MixTrain when training the Cifar10-ResNet18 benchmark using the Adam optimizer. The paper has been revised to include this analysis (Table 4, Sec. 7.2). As can be seen, MixTrain continues to achieve a good accuracy versus speed-up trade-off.  Table 4 in Sec. 7.2 also includes experiments that indicate that changing other training hyper-parameters, i.e., when using different learning rate schedules, does not impact the accuracy versus training speed-ups achieved by our approach.
>
> =============================================================
>
> Re: Applicability to other tasks
>
> In image classification problems, mixing inputs does not hinder the DNN’s ability to recognize the features of the constituent inputs [1,2], as the DNN’s input space is continuous. Thus, the mixed input can be considered to be a new data point in the input space. However, on tasks such as NLP, each word in a sentence is from a discrete dictionary, represented using a token generated using an embedding table. Thus, the new token generated by, for example, linearly averaging the tokens of K constituent sentences, may not exist in the dictionary the network is trained on. Furthermore, networks designed for such tasks factor into consideration the position of each word in a sentence, and their relation to other words. The impact of mixing tokens of different sentences is thus unclear. Consequently, applying MixTrain to such tasks may require significant new research and is beyond the scope of this work.
>
> In semantic segmentation tasks, each pixel in the input image is assigned a target label that indicates the class to which the pixel belongs to. In the context of the MixUp operator, after mixing has been applied, each pixel belongs to two classes, pertaining to each of the constituent inputs. The final target label of the pixel in the composite sample can thus be obtained by averaging the one-hot encoded labels of the corresponding pixel in the constituent inputs. In the context of the CutMix operator, the composite sample contains patches from each of the constituent inputs. Thus, the labels of pixels in a patch of the composite sample can be easily inferred by their value in the original constituent input. However, while such CutMix and MixUp based training for segmentation tasks have been demonstrated in a semi-supervised context, their usage in fully supervised settings has not been widely adopted. Thus, extending MixTrain to such tasks is unclear, and beyond the scope of this work. We have added this discussion to Sec. 7.8 of the paper.
>
> [1]: mixup: Beyond Empirical Risk Minimization. ICLR 2018
>
> [2]: CutMix: Regularization Strategy to Train Strong Classifiers with Localizable Features. ICCV 2019

---

### Author Response · Authors · 2023-07-17
**General Response**

We appreciate and thank all reviewers for their insightful comments about our work. We were wondering if our rebuttal addressed all concerns. If not, we will be happy to address them in subsequent responses and revisions.

---

### Decision · Action_Editors · 2023-07-21

**Recommendation:** Reject

**Comment:**

The paper contains a collection of nice ideas, and a good motivation and analysis. We thank the authors for their engagement in the review process, and submitting this interesting paper.

However, for the reasons outlined above, the current version does not meet TMLR's criterion for acceptance in terms of support for evidence.

Since major revision is not an option, the recommendation is Reject. However, we encourage the authors to consider major revisions: both in the claims on the generality of the method, and substantial expansion of the empirical evidence that the technique can benefit over the results already presented in the MixUp and CutMix papers.

**Audience:**

There would very likely be interest in these techniques by the community. Accelerating NN training is a highly sought after goal, and the ideas presented in the paper are certainly interesting.

**Claims And Evidence:**

The paper presents various techniques to build upon MixUp and CutMix in order to accelerate training. The ideas presented are interesting, and accompanied by good motivation and analysis sections. Overall, the reviews were mixed, while there is some evidence to support the claims of the paper, some reviewers pointed out areas where there is a lack of convincing and clear evidence to support all of the claims.

The two main areas of concern are:

1) The experimental setting is very limited. The main experiment is ResNet50 trained on ImageNet, therefore the baseline setting is almost 8 years old. For example, the CutMix paper (which this method builds on top of) presents results results with a ResNet50 on ImageNet that are 3% higher than any method presented here. The other experiment is on Cifar10 using ViT, which is not a convincing setting to test a technique for accelerating training on large datasets, since performance in this setting is mostly dependent on the regularization/augmentation strategies. Further, the benefits of mixTrain-MixUp seem to disappear completely when compared to random batch skipping, and mostly disppear for mixTrain-CutMix (Fig. 11 (a)).

2) The paper claims a general method for accelerating DNN training, yet evaluates only on image classification (and in a limited setting, see point 1). For reference, the MixUp paper evaluates on image classification, speech, tabular data, image generation, and under adversarial settings. CutMix evaluates on classification, detection, captioning, and robustness. In the discussion, the authors mention that extension to segmentation or NLP may not work. Therefore, the framing as a general acceleration technique is not supported by the evidence.

As a minor point, related work like [DataMUX](https://openreview.net/forum?id=UdgtTVTdswg) should be discussed.


**Resubmission Of Major Revision:**

The authors may consider submitting a major revision at a later time.